# Agonism of Prostaglandin E2 Receptor 4 Ameliorates Tubulointerstitial Injury in Nephrotoxic Serum Nephritis in Mice

**DOI:** 10.3390/jcm10040832

**Published:** 2021-02-18

**Authors:** Ida Aringer, Katharina Artinger, Corinna Schabhüttl, Thomas Bärnthaler, Agnes A. Mooslechner, Andrijana Kirsch, Marion Pollheimer, Philipp Eller, Alexander R. Rosenkranz, Akos Heinemann, Kathrin Eller

**Affiliations:** 1Clinical Division of Nephrology, Department of Internal Medicine, Medical University of Graz, 8036 Graz, Austria; ida.aringer@stpoelten.lknoe.at (I.A.); katharina.artinger@medunigraz.at (K.A.); corinna.schabhuettl@medunigraz.at (C.S.); agnes.mooslechner@medunigraz.at (A.A.M.); alexander.rosenkranz@medunigraz.at (A.R.R.); 2Clinical Department of Internal Medicine 1, University Hospital St. Poelten, 3100 St. Poelten, Austria; 3Otto Loewi Research Center, Division of Pharmacology, Medical University of Graz, BioTechMed Graz, 8036 Graz, Austria; thomas.baernthaler@medunigraz.at (T.B.); akos.heinemann@medunigraz.at (A.H.); 4Clinical Department of Phoniatrics, Medical University of Graz, 8036 Graz, Austria; andrijana.kirsch@medunigraz.at; 5Institute of Pathology, Medical University of Graz, 8036 Graz, Austria; marion.pollheimer@medunigraz.at; 6Intensive Care Unit, Department of Internal Medicine, Medical University of Graz, 8036 Graz, Austria; philipp.eller@medunigraz.at

**Keywords:** EP4 receptor, PGE2, NTS, glomerulonephritis, lipid mediator

## Abstract

Selectively targeting the E-type prostanoid receptor 4 (EP4) might be a new therapeutic option in the treatment of glomerulonephritis (GN), since the EP4 receptor is expressed on different immune cells, resident kidney cells, and endothelial cells, which are all involved in the pathogenesis of immune-complex GN. This study aimed to evaluate the therapeutic potential and to understand the mode of action of EP4 agonist in immune-complex GN using the murine model of nephrotoxic serum nephritis (NTS). In vivo, NTS mice were treated two times daily with two different doses of an EP4 agonist ONO AE1-329 or vehicle for 14 days total. The effect of PGE2 and EP4 agonism and antagonism was tested on murine distal convoluted tubular epithelial cells (DCT) in vitro. In vivo, the higher dose of the EP4 agonist led to an improved NTS phenotype, including a reduced tubular injury score and reduced neutrophil gelatinase-associated lipocalin (NGAL) and blood urea nitrogen (BUN) levels. EP4 agonist treatment caused decreased CD4^+^ T cell infiltration into the kidney and increased proliferative capacity of tubular cells. Injection of the EP4 agonist resulted in dose-dependent vasodilation and hypotensive episodes. The low-dose EP4 agonist treatment resulted in less pronounced episodes of hypotension. In vitro, EP4 agonism resulted in cAMP production and increased distal convoluted tubular (DCT) proliferation. Taken together, EP4 agonism improved the NTS phenotype by various mechanisms, including reduced blood pressure, decreased CD4^+^ T cell infiltration, and a direct effect on tubular cells leading to increased proliferation probably by increasing cAMP levels.

## 1. Introduction

The lipid molecule prostaglandin (PG)E2 mediates pro- and anti-inflammatory effects depending on its local concentration and on the cell type and receptors involved [1,2]. Due to its short half-life, PGE2 is supposed to act locally at the site of production. Prostaglandins originate from arachidonic acid (AA) that is converted to the endoperoxide prostaglandin G2 (PGG2) and further reduced to prostaglandin H2 (PGH2) by the cyclooxygenases 1 (COX1) or 2 (COX2). The different PG subtypes originate from the intermediate PGH2. These subtypes include the prostanoids: Prostaglandin E2 (PGE2), prostacyclin (PGI2), prostaglandin F2α (PGF2α), prostaglandin D2 (PGD2), and thromboxane A2 (TXA2) [1,3,4].

The ubiquitously occurring PGE2 is linked to mediation of pain, fever, inflammation, the regulation of blood pressure, renal perfusion, angiogenesis, and tumor growth. Various functions of PGE2 are mediated by the four different G-coupled receptors, namely EP1–4. The different EP receptors show differential patterns of tissue distribution and signaling pathways [1].

The Gαs-coupled EP4 receptor is found in the gastrointestinal tract, uterus, hematopoietic tissues, endothelial cells, and skin, but can also be detected in the kidney and different immune cells [5,6,7]. The latter make it an attractive target in kidney diseases since specifically targeting EP receptors, such as EP4, might lead to more specific effects compared to drugs, which target the biosynthesis of PGE2 or prostaglandin production in general and thereby have a plethora of side effects including renal toxicity [1,3]. Each prostanoid receptor preferentially binds one of the natural prostaglandins and is named after the one with the highest affinity. Depending on the kidney cell type, various EP receptors are expressed to a greater or lesser degree and determine the overall cellular response to PGE2 [8]. We used the EP4 agonist ONO AE1-329 in our studies. ONO AE1-329, the EP4 receptor–selective compound, exhibits high affinity (Ki = 10 nM) for the EP4 receptor, moderate affinity (Ki = 2100 and 1200 nM) for the EP2 and EP3 receptor, respectively, and low affinity (Ki > 3.3 × 103 nM) for the other prostanoid receptors (EP1, DP, FP, IP, TP) [9].

An attractive research model to study inflammatory driven kidney diseases is the murine model of nephrotoxic serum nephritis (NTS) [10]. It is induced in mice by immunocomplexes, which are deposited on the glomerular basement membrane (GBM), leading to proteinuria and a proliferative form of glomerulonephritis which best resembles membranoproliferative glomerulonephritis (GN). NTS is dependent on TH1 and TH17 cells, which recruit macrophages and neutrophils, respectively, to accelerate disease activity. The disease is limited by regulatory T cell populations, which suppress disease in regional draining lymph nodes, but also locally within the kidney [10,11].

The EP4 receptor is expressed in almost all immune cells, namely B and T cells, NK cells, DCs, neutrophil granulocytes, eosinophils, monocytes, and macrophages [2,12]. The anti-inflammatory actions of PGE2 on macrophages and neutrophil granulocytes have mainly been attributed to EP4 receptor-mediated signaling [13]. On the other hand, the EP4 receptor can mediate the activation of TH17 cells and the increase of the expression of C-C chemokin receptor 7 (CCR7), thereby facilitating the recruitment of T cells [1,14]. CCR7 plays an important role in Treg recruitment to lymph nodes in NTS [15].

Many studies have shown the great impact of PGE2 on kidney physiology. The regulation of parameters, such as renal perfusion, diuresis, sodium excretion, and the regulation of blood pressure by activation of the renin-angiotensin-aldosterone system, is strongly influenced by prostaglandins [1,7,16,17,18,19]. Several studies have demonstrated that the EP4 receptor is expressed in renin-secreting juxtaglomerular granular cells [19,20,21,22]. Thus, the EP4 agonism leads to a decrease in blood pressure by blocking the renin-angiotensin-aldosterone system (RAAS) and by acting vasodilative via cAMP, protein kinase A (PKA), and endothelial nitric oxide synthase (eNOS) [23,24,25]. Furthermore, the EP4 receptor has been described to promote cell survival and to increase proliferation in epithelial cells via an intracellular increase of cyclic adenosine monophosphate (cAMP) levels [26,27,28].

There is evidence that targeting EP4 in this disease model influences disease outcome. Our group recently showed that the EP4 antagonism protects mice from NTS by downregulation of chemokine (C-X-C motif) ligand (CXCL)-5 in tubular cells, thereby inhibiting the recruitment of neutrophil granulocytes to the kidney [29]. Interestingly, positive effects of EP4 agonists on the kidney phenotype have been described in the past without providing clear evidence of the exact mechanism [30]. Thus, the aim of this study was to evaluate whether the EP4 agonist ONO AE1-329 influences the phenotype of NTS and to focus on possible mechanisms such as vasodilatation, proliferation, and immunosuppressive effects.

## 2. Materials and Methods

### 2.1. Animals and Study Design

In this study, 8- to 10-week-old male C57Bl/6J mice (Charles River Laboratories, Sulzfeld, Germany) were immunized s.c. with 2 mg/mL rabbit IgG (Jackson ImmunoResearch Laboratories, PA, USA), which was dissolved in incomplete Freund’s adjuvant (Sigma-Aldrich, St. Louis, MO, USA) with desiccated, nonviable mycobacterium tuberculosis H37a (Difco Laboratories, MI, USA). Three days later, heat-inactivated rabbit anti-mouse GBM serum was injected intravenously. On the same day that GBM serum was injected treatment with high-dose (25 µg/mouse/day equivalent to 1000 µg/kg/day) or low-dose (7 µg/mouse/day equivalent to 280 µg/kg/day) EP4 receptor agonist ONO AE2-329 (ONO Pharmaceutical) or with the same dose of EP4 receptor agonist L-902,688 (Cayman Chemicals, Ann Arbor, Michigan, USA) or vehicle s.c. (12.5% dimethyl sulfoxide dissolved in PBS) twice daily. Mice were sacrificed 14 days after NTS induction and organs were harvested. 

### 2.2. Study Approval

Animal care and experimental procedures were approved by the Austrian Federal Ministry of Science, Research and Economy (BMWFW GZ:66.010/042- WF/ V/3b/2015) and conformed with the Directive 2010/63/EU. Studies are reported in compliance with the ARRIVE guidelines [31,32].

### 2.3. Urine Albumin Quantification by ELISA and Urinary Creatinine Detection

Urinary albumin/creatinine ratio was determined using an albumin ELISA (Bethyl Laboratories, MA, USA) and normalized to urinary creatinine, which was evaluated photometrically using a picric acid-based kit (Sigma-Aldrich). 

### 2.4. Serum Lipocalin-2/NGAL ELISA

To evaluate the serum Lipocalin-2/neutrophil gelatinase-associated lipocalin (NGAL) levels, the Mouse ELISA DuoSet kit (R&D Systems, Park Abingdon, UK) was used according to manufacturer’s instructions.

### 2.5. Assessment of Serum Blood Urea Nitrogen (BUN)

Serum-BUN was evaluated using a commercially available BUN Colometric Detection Kit (Invitrogen, Carlsbad, CA, USA). 

Periodic acid Schiff (PAS) staining harvested NTS kidneys were fixed in 10% neutral-buffered formalin for 24 h and embedded in paraffin afterward. Kidneys were cut in 4 µm sections followed by PAS staining (Merck KGaA, Darmstadt, Germany). For the evaluation of crescents, a minimum of 50 glomerular cross-sections were evaluated. PAS-positive material within glomeruli was scored according to a semiquantitative scoring system as published recently [29].

### 2.6. Immunohistochemistry Staining

Cell proliferations in the different glomerular compartments were assessed as follows: Mesangial hypercellularity was classified as mild (score 1: 4–5 cells/mesangial area), moderate (score 2: 5–6 cells/mesangial area), and severe (score 3: >6 cells/mesangial area). Endocapillary hypercellularity was subclassified as mild (score 1: Present in a single glomerula), moderate (score 2: <50% affected glomerula), and severe (score 3: >50% affected glomerula) [29]. Signs of tubular injury include tubular dilatation, thinning and denudation of the tubular epithelial cells, loss of brush border, and the presence of cellular debris within the tubular lumen, which was scored as described previously [15].

Proliferating-cell nuclear antigen (PCNA) was stained on 4 µm paraffin sections. A three-layer immunoperoxidase staining protocol was used. Antigen retrieval was performed by an automated decloaking chamber for 30 min at 120°C and 20 s at 80°C (Biocare Medical, CA, USA). Tissue was stained with a mouse anti-mouse PCNA antibody (BioLegend, San Diego, CA, USA) with a M.O.M. staining kit (Vector Labs, Burlingame, CA, USA). Positive-stained cells were counted per 6 high-power fields. 

Kidneys were snap-frozen with Tissue-Tek^®^ OCT™ Compound (Sakura Finetek Europe B.V., Netherlands). Then, 4 µm-thick snap-frozen kidney sections were stained using rat-derived primary antibodies for CD4^+^ (*clone* YTS191.1, BioRad, California, SA), CD8+ (*clone* KT15, Serotec), CD68^+^ (*clone* FA-11, BioRad), and anti-Ly6G/C (Gr-1) (*clone* NIMP-R14, Abcam) and biotin-conjugated goat anti-rat IgG (Jackson ImmunoResearch Laboratories) as a secondary antibody. All primary antibodies were diluted 1:200, and the secondary antibody was diluted 1:500. Positive CD4^+^ and CD8^+^ cells, as well as interstitial-infiltrating Ly6^+^ cells, were counted using 6 high-power fields. A semiquantitative scoring system for CD68^+^ positive cells was performed as follows: 0 = 0 to 4 cells stained positive, 1+= 5 to 10 cells, 2+= 11 to 50 cells, 3+= 51 to 200 cells, and 4+= >201 cells stained positive per low-power field. Intraglomerular Ly6 positive stained cells were counted in 50 glomeruli per mouse [29].

### 2.7. Immunofluorescence Staining

To quantify and evaluate renal deposition of autologous and heterologous IgG, as well as C3, cryosection were incubated with serial dilutions of fluorescein isothiocyanate (FITC)-conjugated goat anti-mouse IgG (Jackson ImmunoResearch Laboratories) and FITC-conjugated goat anti-rabbit IgG (Jackson Immuno-Research Laboratories). Semiquantitative evaluation was assessed in a blinded fashion. To evaluate C3 deposition in the kidneys, kidney sections were incubated with serial dilutions of FITC-conjugated goat anti-mouse complement C3 (MP Biomedicals, Eschwege, Germany). 

### 2.8. IgG ELISA

Rabbit IgG (100 µg/mL; Jackson ImmunoResearch Laboratories) was coated over-night on a 96-well plate, followed by incubation with murine serum in serial-doubling dilutions. To identify circulating mouse anti-rabbit immunoglobulin, horseradish peroxidase-conjugated gout anti-mouse IgG (Dako, Carpinteria, CA, USA) was used.

### 2.9. Blood Pressure Measurement

Blood pressure was measured every day for 14 days total via a tail-cuff in a quiet room. Blood pressure measurements were performed by the same investigator using a Kent Scientific Corporation CODA Non-Invasive Blood Pressure System (Torrington, CT, USA). Animals were placed on a heating unit, and the measurement was only started when the tail skin temperature reached 35°C. Each measurement consisted of 5 acclimations and 10 experimental measurements. The mean arterial pressure (MAP) was calculated with accepted values as follows: MAP = ((2xdiastolic pressure) + systolic pressure)/3).

### 2.10. RNA Isolation, Reverse Transcription (RT) Real-Time Polymerase Chain Reaction (PCR)

TRI Reagent^®^ (Sigma-Aldrich) was used for RNA isolation. Complementary DNA (cDNA) transcripts from RNA were synthesized using Superscript III Transcription Kit (Invitrogen, CA, USA), and random primers (Roche, Basel, Switzerland) were used for reverse transcription. Real-time polymerase chain reaction (PCR) was performed on a CFX96 Real-Time System (BioRad, CA, USA). For quantification of respective genes, TaqMan gene expression assays (Applied Biosystems, DA, USA) for *FoxP3*: Mm00475162_m1, *Tnf-alpha*: Mm00443258_m1, *Il10*: Mm00439616_mL, *Tbx*: Mm00450960_m1, *Ifn-gamma*: Mm00801778_m1, *Il17a*: Mm00439619_m1, and *Gata-3*: Mm00484683_m1 were used. Real-time PCR for EP receptor expression was performed using Advanced™ Universal SYBR^®^ Green Supermix with PrimePCR™ SYBR^®^ Green Assay primers for *PTGER1–4* or *GAPDH*, human (all Biorad). SYBR Green Mastermix (BioRad) was used for the detection of *Hprt* or *β-Actin* using the following primers, respectively: Forward 5′GCT TCC TCC TCA GAC CGC TTT TTG C 3′, reverse 5′ATC GCT AAT CAC GAC GCT GGG ACT G 3′, forward 5′GAA GTG TGA CGT TGA CAT CCG 3′, and reverse 5′TGC TGA TCC ACA TCT GCT GGA 3′. *Hprt* and *ß-Actin* were used as reference genes and housekeeping genes for the kidney tissue. The delta-delta Ct(2^-ΔΔCT^) method was used.

### 2.11. Cell Culture Experiments

The distal convoluted tubular (DCT) cell line was immortalized from mice and functionally characterized by Gesek et al. [33]. The DCT cell line was kindly provided by Miriam Banas, (University of Regensburg, Germany).

DCT cells were grown in DMEM/Ham’s F-12 media (Gibco, life technology) supplemented with 5% heat-inactivated FCS (Gibco) and an antibiotic mixture of 1% penicillin/streptomycin and 100 mg neomycin/100 mL (Gibco, life technology) in a humidified atmosphere of 95% air and 5% CO_2_. Then, 4 × 10^3^ or 1 × 10^3^ DCT cells were seeded and grown confluent. Cells were serum- and glutamine-starved for 72 h and treated twice daily with either PGE2 (30–300 nmol/L) (Cayman), EP4 antagonist, ONO AE3-208 (250–1000 nmol/L) or the selective EP4 agonist, ONO AE1-329 (30–300 nmol/L), for 72 h. The supernatant was collected and stored at −70 °C for further analysis. 

### 2.12. Cell Cycle Determination

Cells were fixed, permeabilized, and incubated with propidium iodide (Sigma-Aldrich), Triton X-100 (0.1%) (Fluka Analytical), and Ribonuclease I (Sigma-Aldrich), followed by incubation at 37°C for 40 min. The stained cells were analyzed via flow cytometry on a FACS Calibur flow cytometer (BD Biosciences). The percentage of cells in each cell cycle phase was analyzed with the analyzing software FLOW JO^®^.

### 2.13. Proliferation Assay

DCT cells were seeded, grown to confluence, and treated as described before. Proliferation of cells was performed using MTS (3-(4,5-dimethylthiazol-2-yl)-5-(3-carboxymethoxyphenyl)-2-(4-sulfophenyl)-2H-tetrazolium; Promega, Madison, WI, USA)). Absorbance measurements were performed on a plate reader FLUOstar Omega Spektralphotometer (BMG LABTECH GmbH, Ortenberg, Deutschland) at 490–500 nm.

### 2.14. cAMP Enzyme Immunoassay

Intracellular cAMP in lysed DCT was measured using a competitive enzyme immunoassay system (GE Healthcare Europe, Vienna, Austria). 

### 2.15. Radioimmunoassay (RIA)

A radioimmunoassay for the detection of PGE2 in DCT cells was performed as previously described [34].

### 2.16. Statistical Analysis

Data are shown either as mean ± standard error of the mean (SEM) or mean with raw data. Statistical evaluations were performed using one-way ANOVA followed by Dunnett’s test against vehicle as the control group. Mean arterial pressure was compared for each day using one-way ANOVA for repeated measurements. When comparing only two groups, Student’s t-test or the Mann–Whitney U test was used. Tests were performed depending on the distribution of the data tested by the Kolmogorov–Smirnov test. Fisher’s exact test was used to compare scores. *p*-values of less than 0.05 were considered statistically significant. Statistical analyses and graphs were performed using GraphPad Prism version 5.05 (GraphPad Software, La Jolla, CA) or the statistical software R version 3.3.1. (R Foundation).

## 3. Results

### 3.1. EP4 Agonism Mainly Improves Tubular Pathologies in NTS

On the day of NTS injection, mice were treated twice daily every 12 h with two different doses of EP4 agonist and vehicle. Fourteen days after vehicle-treated mice showed histological features of NTS, such as PAS-positive deposits in the glomeruli, occasional crescent formations, and interstitial immune cell infiltrates (Figure 1A). Whereas low doses of the EP4 agonist histologically resembled vehicle-treated mice, high doses of the EP4 agonist resulted in an overall improved histological phenotype (Figure 1A). Interestingly, high doses of the EP4 agonist mainly resulted in an improved tubular phenotype, as shown by reduced tubular injury score (Figure 1G) and BUN levels (Figure 1K). However, crescents (Figure 1B) remained mainly unchanged, and high-dose EP4 treatment even resulted in increased albuminuria levels 14 days after NTS induction, while no differences were observed 7 days after NTS (Figure 1E,F). Serum-NGAL levels were slightly decreased in the high-dose EP4 agonist group, but significance was not reached (Figure 1J). Low doses of the EP4 agonist improved tubular pathologies (Figure 1G), but not to the extent of high doses of the EP4 agonist. In contrast, glomerular pathologies such as mesangial (Figure 1C) and endocapillary proliferation (Figure 1D) were also improved in mice treated with low doses of the EP4 agonist. Of note as proof of concept, both doses of the EP4 agonist resulted in an increased venous dilatation score (Figure 1I), reflecting their vasodilative potential. This was also reflected by decreased mean arterial pressure (MAP) (Figure 2) after the injection of the high and low doses of the EP4 agonist as evaluated by tail-cuff measurements. In the high-dose EP4 agonist group, MAP dropped below the lower threshold of detection after the day 4 of treatment (Figure 2). Thirty minutes after injection, MAP was again detectable in the high-dose EP4 agonist group but was significantly lower compared to vehicle-treated mice. A significant decrease in MAP was observed in low-dose agonist-treated mice.

Of note, we performed experiments using another EP4 agonist, L-902,688 (Cayman Chemicals, Ann Arbor, Michigan, USA), in NTS mice showing comparable phenotypes (Appendix A).

IgG production is essential for the NTS phenotype. Therefore, to rule out whether EP4 agonist treatment has an impact on anti-rabbit IgG production, we performed an IgG ELISA of mouse serum. Mouse anti-rabbit IgG titers did not differ between the groups 14 days after NTS induction (Appendix A). Furthermore, we did not detect a difference in deposition of mouse anti-rabbit IgG and rabbit anti-mouse IgG on GBM (Appendix A), as well as glomerular C3 deposition between the different groups (Appendix A).

### 3.2. EP4 Agonism Reduces Renal CD4^+^ T-Cell and Neutrophil Infiltration in NTS

Fourteen days after induction of NTS, a significant decrease in the infiltration of CD4^+^ T cells into the kidneys of EP4 agonist-treated mice was observed compared to vehicle-treated mice (Figure 3A). No significant difference was detected for the infiltration of CD8^+^ T cells and CD68^+^ macrophages in immunohistochemistry (Figure 3B,C). Whereas there was no difference in Ly6G^+^ neutrophil cell counts in the glomeruli (Figure 3D), this cell population was significantly decreased in the interstitium of mice treated with high doses of the EP4 agonist (Figure 3E). This effect was not seen in the low-dose EP4 agonist-treated group (Figure 3E). Th1-related cytokines, such as *Tnf-alpha*, *Il-10*, and *Ifn-gamma*, and the Th1 master gene regulator *Tbet* were not differentially regulated on mRNA level in the kidney of the three groups (Appendix A). Interestingly, the master gene regulator of Th2 *Gata3* was significantly decreased in kidneys of mice treated with high doses of the EP4 agonist (Figure 3F). *FoxP3* mRNA in kidneys was also decreased in high-dose EP4 agonist-treated mice 14 days after NTS, but significance was not reached (Figure 3G). Since *Il-17* mRNA remained undetectable in the kidney, lymph nodes were evaluated for the expression of *Il-17*. *Il-17* mRNA expression was reduced in lymph nodes of high-dose EP4 agonist-treated mice, but again these results were not statistically significant (Figure 3H).

### 3.3. EP4 Agonism Increases Proliferation of Tubular Cells In Vivo and In Vitro

EP4 activation has been described to increase cell proliferation and viability [1]. We hypothesized that increased tubular cell proliferation via improvement of tubular regeneration might be an explanation for the improved NTS phenotype in EP4 agonist-treated mice. To evaluate this concept in vitro, DCT cells were treated with EP4 agonist, antagonist, or PGE2 under starving conditions for 72 h. We detected increased concentration-dependent cell viability when cells were treated with PGE2 or the EP4 agonist (Figure 4A,B), whereas incubation with the EP4 antagonist had no effect (Figure 4C). Coincubation of PGE2 or the EP4 agonist with the EP4 antagonist abolished the increased viability of cells, which returned to control levels (Figure 4D,E). Cell cycle analysis by propidium iodide (PI) staining revealed that PGE2, as well as the EP4 agonist, reduced the proportion of late-apoptotic cells seen in the sub-G1 phase (Figure 5A,B), whereas the EP4 antagonist had no effect (Figure 5C). PGE2, but not the EP4 agonist, significantly increased the portion of cells in the S phase (Figure 5B). In vivo, we observed significantly increased numbers of PCNA-positive, proliferating tubular cells in the high-dose EP4 agonist group 14 days after induction of NTS as compared to the vehicle group (Figure 6). Similar data were observed in mice treated with high doses of the EP4 agonist from Cayman Chemicals (Appendix A). Apoptosis, as measured by cleaved Caspase-3 staining, was only found in very few tubular cells and did not show differences between groups.

Of note, the expression of EP receptors by DCTs has not been investigated yet. Therefore, we determined EP1-4 receptor expression in DCT at the mRNA level via qPCR (Appendix A), and found that DCT cells equally expressed *EP1*, *EP3*, and *EP4* receptors. The EP2 receptor was not detectable in DCT cells. To determine the concentration of PGE2 endogenously produced by the DCT cells, we performed a radioimmunoassay (RIA). Although we could detect endogenous PGE2 production, we found only low amounts of PGE2 (0.05 ng/mL) in the supernatants of DCT cells after 24 h (Appendix A).

### 3.4. EP4 Agonism Increases cAMP Production in Tubular Cells In Vitro

It is known that activation of the Gαs-coupled EP4 receptor leads to an intracellular increase in cyclic adenosine monophosphate (cAMP) levels [35,36,37], which usually results in increased proliferative capacity of cells [1]. In line with previous findings, EP4 agonist and PGE2 treatment of DCT cells resulted in a time-dependent increase in intracellular cAMP levels (Figure 7A,B). Both were inhibited by EP4 antagonist pretreatment (Figure 7A,B). EP4 antagonist treatment alone had no effect on cAMP levels in DCT cells (Figure 7A,B).

## 4. Discussion

In the present study, we provided evidence that EP4 receptor agonism has beneficial dose-dependent effects in an experimental model of glomerulonephritis. The EP4 agonist ONO AE1-329 mainly protected tubular injury in our NTS model by improving the proliferation and thereby regeneration of tubular epithelial cells via cAMP increase.

As highlighted in a recent review by Nasrallah et al., targeting EP4 with an EP4 receptor agonist in different experimental models of kidney diseases has both protective and harmful effects on the outcome [8,36,37]. Our data now add to the observed discrepancies. We recently published that EP4 antagonism protects mice from NTS [29], and we now show that the EP4 agonist ONO AE1-329 has protective effects in NTS mice as well. One of the major and most impressive findings of our tested EP-4 agonists is their vasodilatory potential, leading to pronounced dose-dependent hypotensive episodes after injection. Interestingly, the EP4 antagonist did not induce any alterations in blood pressure [29]. We thus believe that the drop in blood pressure might critically influence the phenotype, whereas this is obviously not true for the EP4 antagonist. Thereby, the EP4 antagonist acts completely differently than the EP4 agonist, possibly leading to discrepant results. Furthermore, the EP4 receptor is expressed on different immune cells as well as resident kidney cells, such as endothelial cells and tubular epithelial cells, including cells of the juxtaglomerular apparatus. Thus, the EP4 receptor agonism, as well as antagonism, results in different effects depending on the cell type [1,8,19]. Another explanation might be the fact that, depending on the kidney cell type, various EP receptors are expressed to a greater or lesser degree, and EP receptors determine the overall cellular response to PGE2 or the EP4 receptor agonist or antagonist [18,37,38]. PGE2 acts on different EP receptors, and therefore has opposing effects, considering that PGE2 can cause both vasodilatory and vasoconstrictor responses and influence renal blood flow and hemodynamics [37,39,40]. The reported effects of the EP4 agonist vs antagonist treatment is dependent on the relative expression of EP receptors on the specific cell type, the presence of inflammation, and therefore, PGE2 release or other hormonal signals, e.g., endothelin, angiotensin II and nitric oxide. Moreover, other prostanoids (PGE1, PGD2, PGF2, PGI2) could potentially interact with the EP4 receptor [7,16,37,41]. Still, there are many open questions to answer this discrepancy. EP4 receptor knockout mice are unfortunately critical in breeding, as only 5% of pups are viable due to a patent ductus arteriosus. The surviving mice have an altered kidney development, and reduced numbers of nephrons in these mice are further limits for the feasibility and applicability of our model [1].

Beneficial effects of high-dose EP4 receptor agonist treatment have previously been observed in this NTS model [30,42]. So far, previous studies have used 25-300µg/kg/day of the EP4 receptor-selective agonist ONO AE1-329 [1]. It must be noted that we used 280 μg/kg/day or 1000 μg/kg/day of the EP4 receptor agonists ONO AE1-329, which is considerably higher than the studies performed so far. The vehicle-treated groups might be influenced in their NTS phenotype due to the dimethyl sulfoxide (DMSO) injected, which has been shown to improve passive Heymann nephritis [43]. Still, in our setup, the EP4 receptor agonism mainly improved tubular rather than glomerular injury. This protective effect seems to be primarily mediated by an increased proliferation and regeneration of tubular epithelial cells.

In a cis-diammine dichloroplatinum-induced rat model of renal failure, Yamamoto et al. showed that PGE2, produced by increased COX-1 and mPGES-1 expression, induced epithelial regeneration via the upregulated EP4 receptor in renal tubular cells. COX-1 plays more important roles than COX- 2 in the affected renal tubules [44]. In line with the results found by Yamamoto et al., we observed an increase in proliferating tubular epithelial cells in the high-dose EP4 receptor agonist the treated group in vivo. Therefore, we evaluated the influence of PGE2 selective EP4 receptor agonism and antagonism on a murine distal tubular epithelial cell line in vitro. EP4 receptor agonism also resulted in increased numbers of viable cells and decreased numbers of cells in the sub-G1 phase (reflecting late-stage apoptotic cells) in vitro [29]. The latter observation was also made when cells were incubated with PGE2. The importance of EP4 receptor signaling in this mechanism was underlined by the fact that coincubation of tubular cells with both PGE2 and a selective EP4 receptor antagonist resulted in equivalent numbers of cells in the sub-G1 phase comparable to vehicle-treated cells. Importantly, we hereby prove the specificity of pharmacological EP4 receptor targeting in tubular epithelial cells.

Of note, albuminuria was found to be increased in the high-dose EP4 agonist-treated group 14 days after NTS induction. This effect might be explained by the fact that PGE2 signaling via the EP4 receptor has been shown to increase podocyte injury, causing proteinuria and glomerular damage [45,46,47]. Further studies are needed to delineate the exact pathomechanistic effects of EP4 agonists on podocytes in NTS.

Furthermore, EP4 agonism acted on the vascular system, leading to recurrent hypotensive episodes after treatment, which is a well-known effect of PGE2 leading to vasodilation via EP4 receptor signaling [21]. The EP4 receptor, which is expressed in the juxtaglomerular apparatus, is also suggested to take part in renin production [19], thereby inhibiting the renin-angiotensin-aldosterone system. The protective effect of EP4 agonism could potentially also be mediated by a sustained decrease in blood pressure or by inducing effects comparable to ischemic preconditioning known from ischemia-reperfusion models [48,49,50]. The effect of the EP4 receptor agonist in NTS was dose-dependent, since low-dose agonist treatment caused less hypotensive episodes and a NTS phenotype comparable to vehicle controls. Burne-Taney and coworkers showed that ischemic preconditioning influences systemic immune cells but did not provide evidence of the exact immune cell subtype responsible for the observed effect [50]. Subsequently, Tregs migrating to the kidney due to ischemic preconditioning were found to protect mice from ischemia reperfusion injury [49]. More recently, evidence was provided that limb ischemic preconditioning reduces systemic Il-17 and thereby protects from cerebral ischemia [48]. In line with previous studies, we showed that high-dose EP4 receptor agonist treatment resulted in decreased infiltration of immune cells into the kidney as well as reduced *Il-17* mRNA in the draining lymph node. Of note, systemic T and B cell numbers were not influenced by high-dose agonist treatment. The picture in the low-dose treatment group concerning decreased immune cell infiltration in the kidney was not as clear, despite an obvious protection when it comes to BUN levels and tubular injury. Thus, we hypothesized that EP4-mediated immune mechanisms are responsible for the protective role of EP4 receptor agonist treatment to a lesser extent. Nevertheless, further studies are needed to unravel the exact mechanisms of EP4 receptor agonist-induced immune suppression in NTS.

In summary, EP4 receptor agonism improved the NTS phenotype mainly by improving tubular injury in mice. This was mediated by an EP- induced increase in cAMP production in tubular epithelial cells, leading to increased tubular proliferation and regeneration.

## Figures and Tables

**Figure 1 jcm-10-00832-f001:**
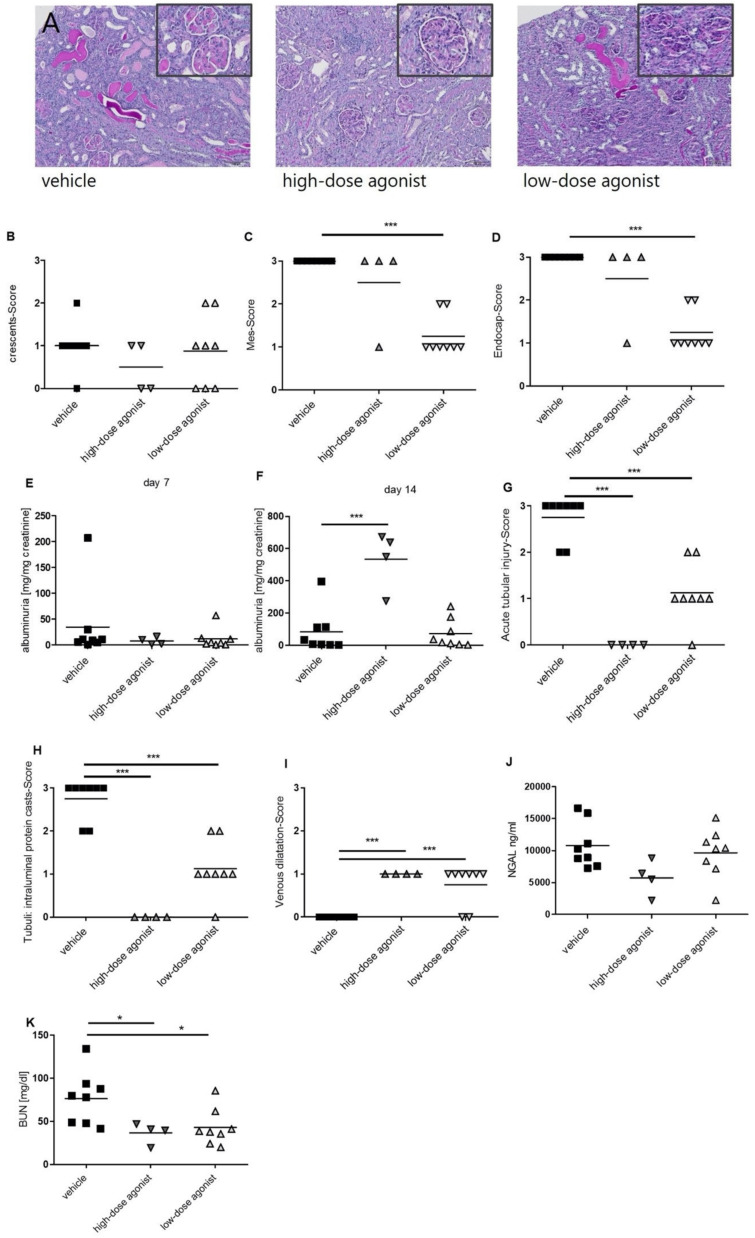
EP4 receptor agonism improved the phenotype of nephrotoxic serum nephritis (NTS). Fourteen days after NTS induction, kidneys of mice treated with vehicle, high-dose EP4 receptor agonist, or low-dose EP4 receptor agonist were harvested, and PAS-staining was performed (**A**). Representative pictures are shown. Stained kidney sections were quantified for glomerular and kidney damage (**B**–**D**). Urine samples collected on day 7 (**E**) and 14 (**F**) were analyzed for albumin and creatinine. In order to quantify for tubular damage acute tubular injury score (**G**) and tubular cast formation, six high-power fields (HPF) (**H**) were evaluated. Venous dilatation score (**I**) was quantified. Serum neutrophil gelatinase-associated lipocalin (NGAL) levels (**J**) and serum blood urea nitrogen (BUN) levels (**K**) were evaluated on day 14. (* *p* ≤ 0.05, *** *p* ≤ 0.001).

**Figure 2 jcm-10-00832-f002:**
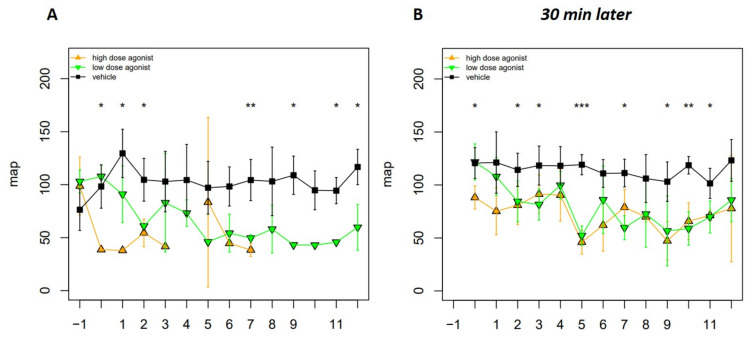
Hypotensive episodes occurred after EP4 receptor agonist administration. Blood pressure was assessed in vehicle- (*n* = 4), EP4 receptor high-dose agonist- (*n* = 4), and EP4 receptor low-dose agonist- (*n* = 4) treated mice after NTS induction. Blood pressure was measured via the tail cuff method immediately (**A**) and 30 min after EP4 agonist injection (**B**) in all three groups. Mean arterial pressure (MAP) is represented as mean ± SEM. Statistical significances are provided compared to vehicle-treated mice (* *p* ≤ 0.05, ** *p* ≤ 0.01, *** *p* ≤ 0.001).

**Figure 3 jcm-10-00832-f003:**
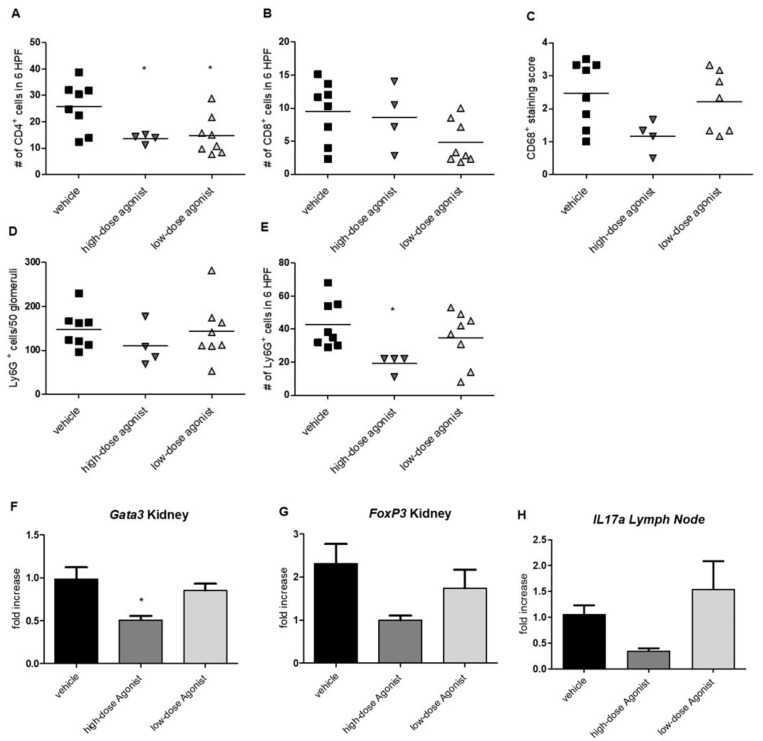
Effects of EP4 agonism on kidney-infiltrating immune cells. Fourteen days after induction of NTS, renal infiltration of CD4^+^ (**A**), CD8^+^ T cells (**B,**) and CD68^+^ macrophages (**C**) were evaluated in mice treated with high- and low-dose EP4 receptor agonist compared to vehicle. Ly6G^+^ (Gr1) cells were evaluated in glomeruli (**D**) and the interstitium (**E**) on day 14 after NTS induction. Fourteen days after NTS induction, quantitative PCR was performed to evaluate *Gata3*, *FoxP3*, and *Il17a* expression in the kidney and lymph node (**F–H**) in mice treated with vehicle (*n* = 6), high-dose EP4 receptor agonist (*n* = 3–4), and low-dose EP4 receptor agonist (*n* = 7–8). Results were compared to the kidneys of healthy mice (*n* = 3). Data are provided (mean ± SEM) as fold increase compared to healthy kidneys. Means are indicated by a horizontal line. Statistical significances are provided compared to vehicle-treated mice (* *p* ≤ 0.05).

**Figure 4 jcm-10-00832-f004:**
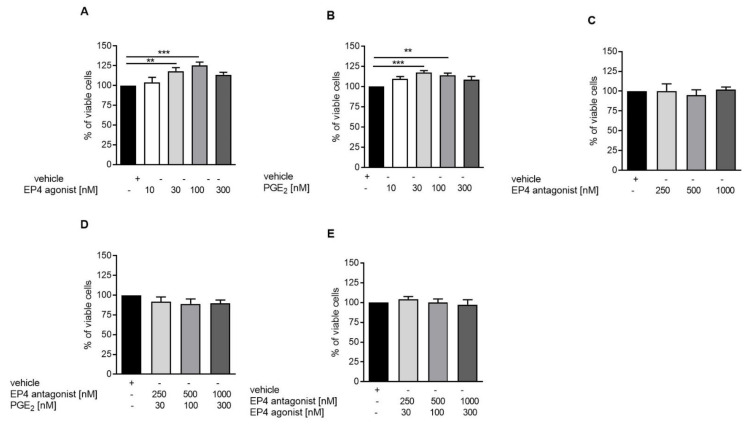
EP4 receptor agonism increases tubular proliferation in vitro. To evaluate proliferative capacity of DCT cells, the MTS test was performed. DCT cells were treated with vehicle, EP4 receptor agonist (**A**), PGE2 (**B**), EP4 receptor antagonist (**C**), and the EP4 receptor antagonist together with PGE2 (**D**), as well as EP4 receptor antagonist together with EP4 receptor agonist (**E**) under starving conditions for 72 h. Data are provided as mean ± SEM from at least five independent experiments. Statistical significances are provided compared to vehicle-treated cells (** *p* ≤ 0.1, *** *p* ≤ 0.001).

**Figure 5 jcm-10-00832-f005:**
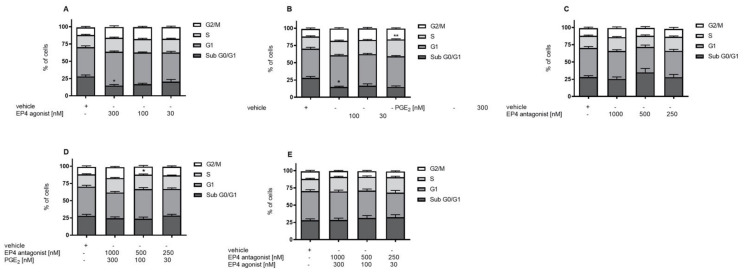
EP4 agonist treatment significantly decreased cells in the sub-G1 phase and increased cells in the G1 phase of DCTs. To evaluate cell viability, a cell-cycle analysis was performed. DCT cells were treated with vehicle (*n* = 7), EP4 receptor agonist (*n* = 6–7) (**A**) PGE2 (*n* = 5–6) (**B**), EP4 receptor antagonist (*n* = 5–6) (**C**), and the EP4 receptor antagonist together with PGE2 (*n* = 6–7) (**D**) or together with the EP4 receptor agonist (*n* = 4) (**E**) under starving conditions for 72 h. Data are shown as mean ± SEM. Statistical significances are provided compared to vehicle (* *p* ≤ 0.05, ** *p* ≤ 0.01).

**Figure 6 jcm-10-00832-f006:**
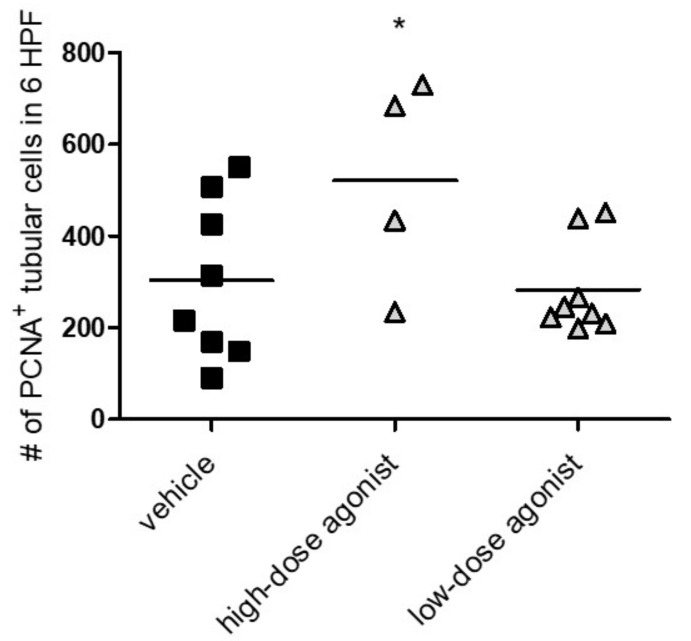
EP4 agonist-treated mice showed increased proliferation of tubular cells in vivo. Fourteen days after NTS induction, mice treated with vehicle, high-dose EP4 receptor agonist or low-dose EP4 receptor agonist were evaluated for PCNA+ tubular cells per six high power fields (HPF). Means are indicated by horizontal lines. Statistical significances are provided compared to vehicle-treated mice (* *p* ≤ 0.05).

**Figure 7 jcm-10-00832-f007:**
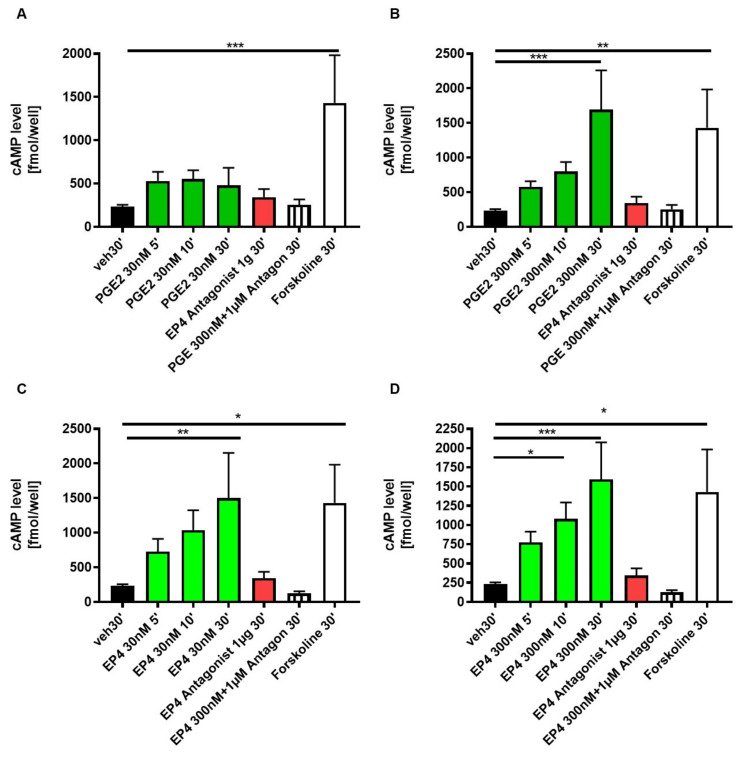
EP4 stimulation of DCTs with ONO AE1-329 (30 nmol/L or 300 nmol/L) significantly increased cAMP production DCTs. Cells were treated with either PGE2 30 nmol/L (**A**) or 300 nmol/L (**B**), EP4 antagonist ONO AE3-208 (1000 nmol/L), or EP4 agonist ONO AE1-329 30 nmol/L (**C**) or 300 nmol/L (**D**). As a control, cells were either not treated (veh) or as a positive control with Forskoline (FSK). Cells were lysed, followed by a 5-, 10- or 30-min incubation before assay. Experiments were performed in duplicates. In total, at least five different experiments were performed. Statistical significances are provided compared to vehicle (* *p* ≤ 0.05, ** *p* ≤ 0.01, *** *p* ≤ 0.001)).

## Data Availability

The data presented in this study are available on request from the corresponding author.

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
