# Peer review of "Agonism of Prostaglandin E2 Receptor 4 Ameliorates Tubulointerstitial Injury in Nephrotoxic Serum Nephritis in Mice"

_jcm, 2021, doi:10.3390/jcm10040832_

Round 1

Reviewer 1 Report

Prostaglandins mediate many pro- and anti-inflammatory functions as well as cell proliferation, regeneration and survival. Their effects depend on type and concentration of the prostaglandin, cell type and receptors involved in signaling. This plethora of functions limits therapeutic strategies targeting prostaglandins in inflammatory or autoimmune disease. Therefore, more specific approaches focusing on the role of single prostaglandin receptors are of great interest, which investigate both blockade and stimulation of particular receptors in disease models.

Prostaglandin E2 (PGE2), which exerts its biologic effects via four distinct E-type prostanoid receptors (EP1 to 4), plays a pathophysiologic role in mediating pain, fever, inflammation and regulates blood pressure, renal perfusion, angiogenesis and tumor growth. Experimental evidence suggests that PGE2 via EP3 (Kvirkvelia N et al. Am J Physiol Ren Physiol. 2013;304:F463) or EP4 receptor activation (Nagamatsu T e al. J Pharmacol Sci. 2006;102:182) has protective effects in murine nephrotoxic serum nephritis, a well-established model of immune complex-mediated glomerulonephritis. This data mainly suggested that PGE2 promoted regeneration of glomerular cells. In contrast, Aringer et al. (the authors of this manuscript) recently showed that EP4 receptor blockade may also be beneficial in the same model, describing reduced tubular Cxcl-5 expression and subsequently decreases interstitial neutrophil accumulation and renal pathology in P4 antagonist-treated NTN mice (Aringer et al. Am J Physiol Ren Physiol. 2018;315:1869-80).

In this manuscript, Aringer et al. now further investigate the apparently discrepant effects of PGE2-mediated EP4 receptor activation in the nephrotoxic serum nephritis model. The authors treated nephritic mice with two doses of an EP4 agonist (ONO AE1-329) or vehicle starting on the day of disease induction and analyzed renal pathology at day 14. Additional in vitro studies were performed in a distal tubular cell line (DCTs). In their in vivo experiments the authors show that the higher dose of the EP4 agonist improved glomerular und tubular injury, reduced interstitial CD4+ T cell infiltration, and increased proliferative capacity of tubular cells. Systemicly, EP4 agonist treatment resulted in vasodilatation and hypotension. In vitro, EP4 stimulation resulted in increased DCT proliferation. The authors conclude that EP4 receptor activation protected from nephritis mainly by improving tubular injury through increased tubular proliferation and regeneration.

This data adds to our understanding of potentially beneficial functions of PGE2-mediated EP4 receptor activation in glomerulonephritis. However, the apparent discrepancies to the protective effects of EP4 receptor blockade in the same model reported in the authors` previous work cannot really be explained, and implications of these discrepancies for potential therapeutic approaches remain unresolved. Cell-specific knockouts of the EP4 receptor in tubular versus glomerular cells may be one approach to target this question, but such experiments may lay well beyond the scope of the current manuscript.

All presented experiments appear to be well performed and analyzed. However, the paper needs a few major improvements to support the authors` conclusions. Moreover, the Methods section needs some corrections.

The authors should address the following points:

Major points:

  1. The main finding of the paper is the improved renal phenotype in the high-dose agonist group, including tubular injury, functional parameters, and leukocyte infiltration. According to the figures this experimental group consisted of only n=4 mice (in contrast to 8 mice in each of the other two groups). This number of animals is too low to allow clear conclusions on the results of the in vivo experiments. In view of the very well established and reproducible nephrotoxic serum nephritis model with its short duration (14 days) the authors must analyze an additional number of mice in this group (at least another set of 4 to 5 mice) and provide a pooled analysis of all data.
  1. For a conclusive demonstration that EP4 agonism increases cAMP levels in DCT cells the authors should include two agonist-treated groups (low- and high-dose) in their in vitro-experiment (Figure 7).

Minor points:

  1. Title: As the key findings of this study relates to improved tubulointerstitial injury this should be reflected in the title of the work, e.g. “Agonism of prostaglandin E2 receptor 4 ameliorates tubulointerstitial injury in nephrotoxic serum nephrtis in mice”.
  1. To readers who are not experts in the field information on receptor specificity of the EP4 agonist ONO AE1-329 in mice and its concentration dependency should be given in the Introduction section, as this is crucial for interpretation of the results of this study.
  1. Methods section, Animals and study design: The authors do not mention the nature of the vehicle used to dissolve ONO AE1-329. This is crucial as some vehicle solutions, especially in formulations for s.c. application of small molecule receptor antagonists or agonists may significantly alter immune responses and renal phenotype in the nephrotoxic serum nepritis model. In this scenario, beneficial effects of EP4 activation would be super-imposed (or dependend ?) on effects of the vehicle, which should be discussed accordingly.
  1. Methods section, Immunohistochemistry staining. Analysis of renal injury in PAS-stained sections is somewhat misplaced here, as this is not immunohistochemistry.
  1. Methods section, Immunohistochemistry staining: It is unclear how mesangial hypercellularity was classified. Were mean values of 50 glomeruli/mouse compared ? Which score was given when a mesangial area contained 5 cells: score 1 or 2 ? How was a score given to a single glomerulus which should contain many mesangial areas ? Many mesangial areas may only be partially detected in a given section leading to underestimation of the cell number/glomerular section. How was this potential error addressed ? Overall, I feel that the method described here was rather a subjective, semiquantitative procedure. More accurately, mesangial cell number per glomerulus should be counted in 50 glomerular cross-sections per mouse.
  1. Methods section, Immunohistochemistry staining: The section lacks any information on how tubular injury scores were obtained (results illustrated in Figure 1G, 1H, 1I). Please provide this information.
  1. Methods section, Immunohistochemistry staining: Please provide information on clone and dilution for antibodies used for immunohistochemistry.
  1. Methods section, Immunohistochemstry staining: I assume an anti-Ly6G (rather than Ly6) antibody was used to detect neutrophils ?
  1. Methods section, Immunofluorescence staining: Please re-phrase “kryo sections” (line 151).
  1. Methods section, Immunofluorescence staining: What is meant by “evaluation of an independent observer” ? I assume the observer was blinded regarding group allocation of analyzed mice. Indeed, the authors should confirm that all histology and immunohistochemistry evaluations were performed in a blinded fashion.
  1. Methods section, Cell culture experiments: More detailed information on the DCT cell line (not just the source) should be provided, including a respective reference. Line 191: Correct to “..1% penicillin/…”.
  1. Methods section, Proliferation assay: Why were confluent, rather than sub-confluent cells used in the proliferation assay ?
  1. Methods section, cAMP enzymic immunoassay: Was really cAMP secretion by DCTs detected? Rather, cAMP concentration following cell lysis should be detected, as intracellular cAMP content is the relevant read-out.
  1. Results section, EP4 agonism improves mainly tubular pathologies in NTS: Increased albuminuria levels 14 days after NTS induction in the high-dose group (Figure 1F) are not explained. The associated hypotension in this group should rather lead to lower albuminuria. In the Discussion section (line 396) the authors speculate that EP4 receptor-mediated podocyte injury would be one explanation. To further explore this, the authors could quantify glomerular podocyte number and integrity by staining sections for WT-1 and nephrin. One would expect lower podocyte numbers or at least less intense membrane staining for nephrin in the high-dose agonist group.
  1. Results section, EP4 agonism improves mainly tubular pathologies in NTS: In Figure 1J no significance value is indicated for data of the high-dose agonist group in comparison to vehicle. However, the authors describe reduced NGAL levels in the Methods section (line 236). Please clarify.
  1. Results section, EP4 agonism reduces renal CD4+ T cell and neutrophil infiltration in NTS: In Figure 3G no significance value is indicated for data of the high-dose agonist group in comparison to vehicle. However, the authors describe ”significantly decreased” Foxp3 (line 269). Please clarify. Similarly, no significance is illustrated for reduced Il17a expression levels in lymph nodes, despite claiming that “Il17 mRNA expression was clearly reduced” in the text.
  1. Results section, EP4 agonism increases proliferation of tubular cells in vivo and in vitro: The authors nicely demonstrate increased proliferative capacity of tubular cells in the high-dose agonist group. It would be interesting whether the number of apoptotic/necrotic tubular cells is also affected in vivo. For this, the authors could quantitate TUNEL-positive tubular cells in the kidney sections.
  1. Discussion section, line 416: The authors discuss that “systemic T and B cell numbers were not influenced by high-dose agonist treatment”. However, this data is not shown. Were lymphocyte numbers evaluated in peripheral blood or spleen ?
  1. Refernces: Reference 13 is cited incorrectly (title, publication year).

Reviewer 2 Report

In the present study, authors have reported that the EP4 agonist 356 ONO AE1-329 protected tubular injury in the NTS model by improving proliferation and thereby regeneration of tubular epithelial cells via cAMP increase. In this regard, this study provides some interesting and important information. However, I have several comments as follows.

1) In the previous report by the authors (Ref. 29), EP4 antagonists were found to inhibit the onset of nephritis, and I have the impression that there are some effects on EP4 antagonists in other research reports. Since the results of the present report are inconsistent with those of previous reports, more explanation and discussion are needed.

2) The mechanism by which EP4 agonists inhibit nephritis is that they improve tubular damage by promoting tubular regeneration, but since NTS is a model of immune complex nephritis, glomerular lesions should also be evaluated.

3) The figures are small and the text is difficult to read. In particular, the pathological photographs in Figure 1 should be larger.

4) In Figure 2, the graph lines for high dose/low dose are the same green color, which is difficult to understand.

Round 2

Reviewer 1 Report

I strongly suggest that the authors include their results obtained with a second EP4 receptor agonist in the current manuscript, at least as Supplemental Figures. Reproducing a very similar phenotype with a second agonist strongly supports the overall conclusion of the paper. Thus, these results should be accessable to readers of the paper, not just to the reviewer.

Author Response

We now include the data testing an alternative EP4 agonist as suggested by the Reviewer as supplemental figure to our manuscript. Additionally, we added respective information to the materials and methods section.

Reviewer 2 Report

The author's adequately addressed my comments in the text.

Author Response

We thank the Reviewer for coming to the conclusion that our paper is now ready to be published.

This manuscript is a resubmission of an earlier submission. The following is a list of the peer review reports and author responses from that submission.